# Self-Supervised Adversarial Training via Diverse Augmented Queries and Self-Supervised Double Perturbation

**Ruize Zhang**
Institute of Computing Technology,
Chinese Academy of Sciences
University of Chinese Academy of Sciences
Beijing, China
zhangruize21b@ict.ac.cn

**Sheng Tang**[*]
Institute of Computing Technology,
Chinese Academy of Sciences
University of Chinese Academy of Sciences
Beijing, China
ts@ict.ac.cn

**Juan Cao**
Institute of Computing Technology,
Chinese Academy of Sciences
University of Chinese Academy of Sciences
Beijing, China
caojuan@ict.ac.cn

## Abstract

Recently, there have been some works studying self-supervised adversarial training, a learning paradigm that learns robust features without labels. While those works have narrowed the performance gap between self-supervised adversarial training (SAT) and supervised adversarial training (supervised AT), a well-established formulation of SAT and its connections with supervised AT are under-explored. Based on a simple SAT benchmark, we find that SAT still faces the problem of large robust generalization gap and degradation on natural samples. We hypothesize this is due to the lack of data complexity and model regularization and propose a method named as DAQ-SDP (Diverse Augmented Queries Self-supervised Double Perturbation). We first challenge the previous conclusion that complex data augmentations degrade robustness in SAT by using diversely augmented samples as queries to guide adversarial training. Inspired by previous works in supervised AT, we then incorporate a self-supervised double perturbation scheme to self-supervised learning (SSL), which promotes robustness transferable to downstream classification. Our work can be seamlessly combined with models pretrained by different SSL frameworks without revising the learning objectives and helps to bridge the gap between SAT and AT. Our method also improves both robust and natural accuracies across different SSL frameworks. Our code is available at https://github.com/rzzhang222/DAQ-SDP.

## 1 Introduction

Deep neural network has shown its power in various machine learning tasks. In spite of its beneficial properties in optimization and generalization, deep neural network is vulnerable to adversarial attack as samples with carefully designed tiny perturbations may cause significant deviations of model

---

[*]Corresponding author

38th Conference on Neural Information Processing Systems (NeurIPS 2024).

predictions [16, 25, 12, 36]. One of the most successful defenses tackling this problem is adversarial training, which generates perturbations that makes the largest output deviation and trains the model with perturbed samples [40, 25]. Many following methods [21, 10, 39, 32, 40] have further developed more advanced adversarial training techniques based on this adversarial training framework.

The adversarial defenses mentioned above require full supervision. However, in real situations full labels may not be available. For the semi-supervised setting, some works [26, 3, 2] suggested that unlabeled data can be useful in improving model robustness by designing auxiliary pseudo label losses. However, the performance is largely affected by the amount of available labels. Later, some works studied adversarial robustness in self-supervised learning (SAT). In this scenario, adversarial training (AT) is integrated with self-supervised learning (SSL) to get robust features that can be efficiently finetuned [22, 20, 14, 38, 18, 24, 37, 23]. Some works [22, 20, 14, 24, 37] combined contrastive learning (CL) with AT to tackle this task. In this paper we term those works as ACL methods.

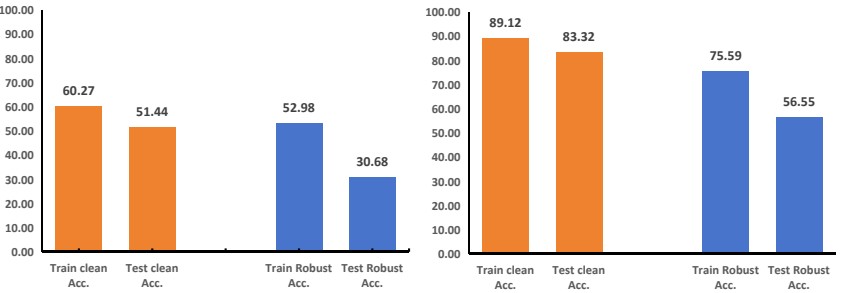

Figure 1: **Motivation:** Large robust generalization gap and reduced clean accuracy for SAT (SimCLR) on CIFAR-100 and CIFAR-10. The left part of the figure is the results on CIFAR-100 and the right part is for CIFAR-10. The robust generalization gap is over 20% on both datasets.

The ACL methods above were restricted to the CL framework and could not generalize to other SSL methods. Previously, Zhang et al. [38] proposed to disentangle the task of SSL and AT into two stages of learning, which first trains SSL models with natural data and then enables AT under the pseudo-supervision of the naturally trained SSL features. This learning framework brings consistent performance boost for both contrastive and non-contrastive methods. Moreover, the feature pseudo-supervision in this framework takes a form similar to the supervision in supervised and semi-supervised AT, thus providing a unified perspective to analyze AT paradigms under different supervision. In this paper, we regard this learning framework as a simple benchmark for SAT.

While previous works on ACL and SAT have achieved remarkable performance that is comparable to supervised AT, there is neither a well-established formulation nor an analysis of the general learning process as in the supervised counterpart. To better understand the limitations of this learning paradigm, we start from the SAT benchmark proposed by Zhang et al. [38], which advocates an SAT process that is much more efficient than previous ACL works (100 vs. 1000 epochs of adversarial training). As shown in Figure 1, we record finetuned train-set and test-set accuracies after SAT pretraining on CIFAR-10 with ResNet-34 as backbone and find that there is a robust generalization gap of around 19% and clean accuracy gap of around 6%. The results are similar for CIFAR-100, which gives a robust generalization gap of around 22% and clean accuracy gap of around 9%. The results show that there exists a large robust generalization gap and natural performance degradation, similar to the AT counterpart.

Given that both SSL and AT are hard tasks [38], we hypothesize these problems to be caused by insufficient data complexity and model regularization. In order to solve the problems, we propose a method termed as DAQ-SDP (Diverse Augmented Queries Self-supervised Double Perturbation) and handle the problems from two aspects. First, while previous papers [24, 38] concluded that complex augmentation techniques are crucial for SSL but harmful for SAT robustness, we argue that strong and diverse augmentations could help SAT if used properly. Specifically, we find that every training distribution is worth one set of BatchNorm layers in SAT and propose an Augmentation-Adversary (Aug-Adv) Pairwise-BatchNorm adversarial training method. It turns out that naturally trained SSL models with a single set of batch norm layers can provide effective guidance for multi-branch adversarial training. Second, while previous works on SAT focused on designing sample adversarial

perturbation for SSL, we find that self-supervised model perturbation also contributes to downstream robustness with a proper training scheme. Despite the task mismatch from pretraining to finetuning, there exists a cross-task transferability of robust generalization. Our work finds a general method that can be directly applied to different SSL frameworks without revising learning objectives, providing insights that contribute to the understanding of SAT.

As far as we know, we are the first to analyze SAT as a general learning paradigm in an SSL-framework-agnostic way and solves its problems by revealing its traits related to clean performance degradation and robust generalization. The main contributions are summarized as follows:

- In contrast to the conclusion of previous works [38, 24], we suggest that strong and diverse augmentations can boost self-supervised robustness and propose an Aug-Adv Pairwise-BatchNorm technique for better robust generalization and less natural degradation.

- Different from previous works [22, 20, 14, 38, 18, 24, 37, 23] that focused on introducing sample adversarial perturbation from supervised AT to SSL, we use model perturbation in SSL pre-training to boost downstream robustness. We then propose a self-supervised double perturbation scheme in the later stage of SAT to improve robust generalization without affecting the learning of natural features.

- We conduct experiments on CIFAR-10 and CIFAR-100, the commonly used datasets in previous works. On CIFAR-100, our proposed method improves over 2% on AutoAttack [7] and clean data results with ResNet-34. On CIFAR-10, our method improves over 1% on AutoAttack [7] and over 2% on clean data results with ResNet-34. The experimental results demonstrate the effectiveness of our method across SSL frameworks, models and datasets.

## 2 Related Works

The task of SAT integrates AT into SSL and aims at learning robust feature representations that enable efficient finetuning. Here we will give a brief summary of previous works related to this field.

**Supervised Adversarial Training (supervised AT)** Given deep neural network's vulnerability to adversarial perturbations, many defense methods have been proposed. Among them adversarial training, originally proposed by Goodfellow et al. [16], has become a prevalent method. Adversarial training simulates a min-max process that finds the perturbation with largest distortion and then minimize the training loss over the adversarial data. Madry et al. [25] proposed a representative adversarial training method and uses random initialized multi-step projected gradient descent to generate adversarial samples. Many following works [21, 10, 39, 32, 40] further revised the adversarial learning framework and applied more advanced techniques such as logits pairing, boundary guidance and consistency regularization for improving robustness while keeping the accuracy on clean data.

**Self-Supervised Learning (SSL)** Self-supervised learning is the task of learning feature representations with no label available. In this case, a discriminative self-supervised method generally relies on a pretext task for pretraining to learn useful information. Previously, many pretext methods have been proposed [15, 11, 28]. One of the most successful pretext task is instance discrimination [6, 4]. Contrastive learning defines a instance discrimination task and learns the features using similarities between positive and negative pairs. The recent development from contrastive learning methods MoCo [6] and SimCLR [4] to non-contrastive methods BYOL [17] and SimSiam [5] has revised the contrastive loss to a positive-pair loss and further simplified the framework. Moreover, some works [29, 13, 19] combined the contrastive framework with some more advanced techniques including positive and negative pair mining, prototypes and customized contrastive view crafting to enrich the information learned by the contrastive framework for effective and efficient learning.

**Self-Supervised Adversarial Training (SAT)** The development of self-supervised learning provides a new direction for acquiring robust features. In self-supervised learning, the emergence of instance discrimination as the new state of art self-supervised pretext task provides a natural setting for adversarial robustness to fit in. While constructing reliable decision boundary using ground truth labels is not feasible, previous works [22, 20, 14, 24, 37] proposed to exploit contrastive loss adversarial training for promoting robustness. The assumption is that if the feature space near the data sample is smooth enough, the feature prediction of adversarially perturbed data will be consistent with its clean counterpart. Gupta et al. [18] suggested that contrastive learning has intrinsic sensitivity to adversarial perturbations and proposed a simple method to remove false negative pairs. This

method enhances robustness in SSL without revising AT and is orthogonal to ours. Xu et al. [37] proposed to use causal reasoning in ACL and used adversarial invariance regularization to enhance ACL. While achieving an impressive performance, the methods above were still restricted to the contrastive learning framework.

Later, Some works have started to explore robustness in the broader SSL picture. Zhang et al. [38] formulated SAT as a two-stage framework that first trains an SSL model and then uses the learned features as guidance for AT. This work has set a strong baseline for this task and can be directly generalized to different SSL frameworks. Thus we take this two-stage SAT framework as the baseline to explore robustness in the broader picture of SSL. Kim et al. [23] proposed an interesting idea that carefully crafted targeted adversarial perturbations can help enhancing robustness for non-contrastive SSL methods. However, the improvements on contrastive frameworks are not as consistent as in the non-contrastive case. Moreover, some works [38, 24] suggested that complex augmentations are crucial for SSL but destructive for SAT. In this work, we approach the task of SAT by studying its learning process and drawing an analogy to supervised AT, with the goal of acquiring a better understanding of the difference and similarity between these two learning paradigms. We then propose a method to tackle the potential problems in SAT. In the following sections, we will introduce the problem statement and then describe our motivation and method.

## 3  Preliminary

**Self-Supervised Learning** As the most representative contrastive learning framework, SimCLR [4] uses data in the same batch as negative pairs and optimizes the following objective:

$$\ell_{CL}(\tau_1(x), \tau_2(x)) = -\log\left(\frac{\exp(\text{sim}(z_i, z_j)/t)}{\exp(\text{sim}(z_i, z_j)/t) + \sum_{k \neq i}^{N} \exp(\text{sim}(z_i, z_k)/t)}\right). \tag{1}$$

In the equation above, $\tau_1(x), \tau_2(x)$ are two augmented views of the same image. $z_i = g \cdot f(\tau_i(x))$ is the projected feature of the corresponding view. $z_i$ and $z_j$ are a positive pair. $N$ is the number of negative samples.

Contrastive SSL relies on large batch size or extra maintained queue for negative pairs and can be computationally expensive. In contrast, positive-only frameworks only include positive pairs. The learning objective of SimSiam [5], a representative positive-only method, can be formulated as:

$$\ell_{ss}(\tau_1(x), \tau_2(x)) = -\frac{1}{2}\frac{p_1 \cdot stopgrad(z_2)}{\|p_1\|_2 \|z_2\|_2} - \frac{1}{2}\frac{p_2 \cdot stopgrad(z_1)}{\|p_2\|_2 \|z_1\|_2}. \tag{2}$$

In the equation above, $\tau_1(x), \tau_2(x)$ are two augmented views of the same image. $z_i = g \cdot f(\tau_i(x))$ and $p_i = h \cdot z_i$ are the projected and predicted feature of the corresponding view, in which g is a projector helps to preserve instance dicriminative features and h is a predictor helps to prevent model callapse.

**Adversarial Contrastive Learning** Based on the framework of contrastive learning (CL), multiple previous works have proposed adversarial contrastive learning (ACL) methods, which aims at improving the robustness of the learned features. Pevious works have adopted such a learning framework but each had some revision of the loss term [22, 20, 14]. In general, those methods can be formulated as:

$$\ell_{CL}^{\text{adv}} = \ell_{CL}(\tau_1(x), \tau_2(x), x^{\text{adv}}), \tag{3}$$

where

$$x^{\text{adv}} = x + \arg\max_{\delta} \ell_{CL}(\tau_1(x), \tau_2(x), x + \delta). \tag{4}$$

In the equations above, $\ell_{CL}^{\text{adv}}$ is the adversarial contrastive loss with an extra variable for the adversarial view. It is often calculated as the average of the pairwise contrastive loss [14]. $x^{\text{adv}}$ is the adversarial sample generated with this loss.

**Self-Supervised Adversarial Training** To find a general method that can improve robustness for different SSL frameworks, we start from a basic SAT framework [38]:

$$\ell_{\text{stage1}} = \ell_{SSL}, \tag{5}$$

and

$$\ell_{\text{stage2}} = \text{Sim}(f_2(x), f_1(x)) + \lambda \cdot \text{Sim}(f_2(x^{\text{adv}}), f_2(x)), \tag{6}$$

where

$$x^{\text{adv}} = x + \arg\max_{\delta}(-\text{Sim}(f_2(x + \delta), f_1(x))). \tag{7}$$

This SAT framework separates the adversarial training process into two stages. In the first stage, an SSL model $f_1$ is trained with clean data. Then the features predicted by the clean model are used as pseudo-supervision for adversarial training of $f_2$ in the second stage. The clean samples providing guiding features can be regarded as queries that help to distill useful information from the clean model. Compared to ACL methods mentioned above, this type of method is more general and also more computationally efficient. In our work, we adopt this framework as the baseline.

## 4 Method

In this paper, we aim at finding a method that can solve the problem of robust generalization and clean accuracy degradation in SAT. Note that our work proposes a general method that can be directly combined with different pretrained SSL models for improvements instead of requiring adversarially re-training the models from scratch. Figure 2 demonstrates the overall framework of our method. In the following sections, we will introduce each part of our method.

### 4.1 Diverse Augmented Query

Previously in supervised AT, some works have discussed about the effects of complex augmentation strategies on robustness [31, 30, 1]. The idea is to fit the model to the labels on the generalized sample distributions to reduce overfitting. However, in SAT, there is no ground truth label to provide supervision on training or generalized data distributions. We hypothesize that SSL models trained with natural data, especially instance discrimination based ones, already contain certain level of generalization capability as the feature space is learned with large amount of data under strong and complex augmentations. However, this capability of generalization can be lost during AT. Thus we propose to use the diversely augmented clean features as supervision for AT.

In the field of ACL and SAT, previous works [38, 24] have concluded that strong and diverse augmentations are essential for SSL but harmful for robustness. Thus Luo et al. [24] proposed to gradually reduce the strength of augmentation during the training and Zhang et al. [38] proposed to only use random resized crop and horizontal flip in adversarial training. In this paper, we argue that diversely augmented samples help SAT given that the model has sufficient capacity, as it's actually essential for the robust model to distill rich information to keep the capability of generalization.

Note that data augmentation strategies including AutoAugment [9] and RandAugment [8] require labels to calculate validation accuracy in the process of searching for optimal augmentation policies, thus can not be directly applied to SAT. In this paper, we propose to use TrivialAugment [27], which is a dataset-independent and search-free method that randomly samples the augmentation policy and strength. Since we need to improve the generalization while also specialize on the testing distribution, we need to fit the adversarial model with clean model features both on the diversely augmented and basic augmented distributions. Inspired by previous work in supervised AT [1], we adopt a multi-stream structure that takes samples from different distributions as inputs. In the previous ACL and SAT works, there were no clear conclusion on the usage of BatchNorm layers. While some works [22, 14] suggested to use different BatchNorm layers for adversarial sample, Zhang et al. [38] suggested that separate BatchNorm layers are not necessary. In this work, we find that each training distribution is worth one set of BatchNorm parameters in SAT and even if the clean model only contains one set of BatchNorm layers, it can provide effective guidance for the multi-BatchNorm adversarial model. Specifically, we define four different input streams based on the combination of their adversarial type (adversarial vs. natural) and augmentation type (strong vs. weak), where the basic augmentation contains random resized crop and horizontal flip while diverse augmentation contains Trivial Augmentation [27] and basic augmentation. In the first stage

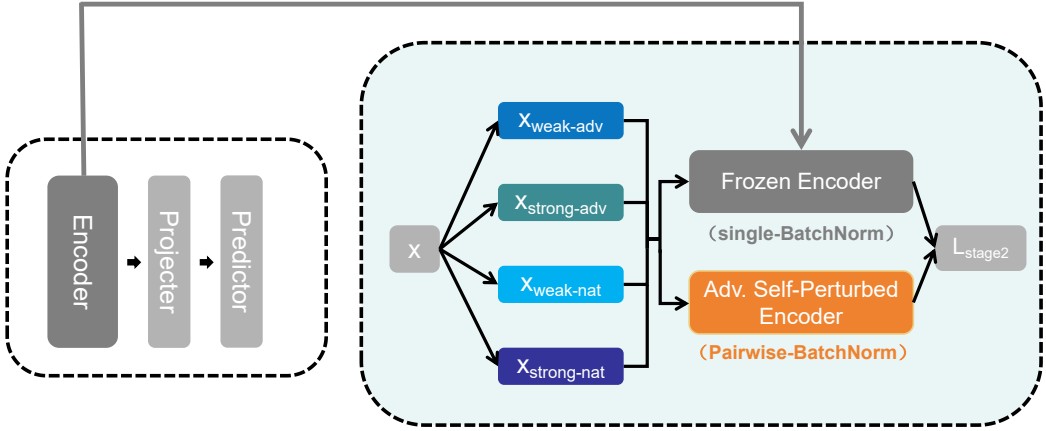

Stage1：Self-Supervised Learning     Stage2：Adversarial Training

Figure 2: A demonstration of our proposed DAQ-SDP. The single BatchNorm encoder of pretrained model is extracted as teacher for our Pairwise-BatchNorm robust encoder for adversarial training.

of our method, we can either use a single set of BatchNorm to train a clean model or directly use a pretrained SSL model with no consideration for robustness. Then in the second stage, we propose an Aug-Adv pairwise-BatchNorm strategy for adversarial training, where each stream of features is pseudo-supervised by the features predicted by the clean model. After training, we only keep the basic-adv BatchNorm layers.

The formulation of our method is:

$$\ell_{diverse-aug} = \ell_{clean} + \lambda \cdot \ell_{adv}, \tag{8}$$

where

$$\ell_{clean} = \sum_{aug_i} \text{Sim}(f_{2-aug_i-clean}(x_{\text{aug}_i}), f_1(x_{\text{aug}_i})), \tag{9}$$

and

$$\ell_{adv} = \text{Sim}(f_{2-aug_i-adv}(x_{\text{aug}_i}^{\text{adv}}), f_{2-aug_i-clean}(x_{\text{aug}_i})), \tag{10}$$

and

$$x_{\text{aug}_i}^{\text{adv}} = x + \arg\max_{\delta}(-\text{Sim}(f_2(x_{\text{aug}_i}^{\text{adv}}), f_1(x_{\text{aug}_i}))). \tag{11}$$

In the equations above, $\text{Sim}(.,.)$ is the cosine similarity between two features and $aug_i$ corresponds to the augmentation type mentioned above. $f_{2-aug_i-adv}$ and $f_{2-aug_i-clean}$ are the student model with the corresponding pairwise-BatchNorm layers. Our method forces the pseudo-supervised adversarial model to inherit the generalization capability from clean model with diversely augmented queries. The AugAdv pairwise-BN helps features of each training distribution to fit the clean feature counterparts without interfering with each other. The experimental results demonstrate that diverse and complex augmentations can improve SAT robustness. This finding helps to narrow the gap between improving SAT and supervised AT.

## 4.2   Adversarial Self-Perturbed Weight

Previous works in ACL and SAT have borrowed the idea of adversarial sample perturbation from supervised AT to SAT and made revisions either on the specific training loss term [20, 22] or the effective way of generating sample perturbations [14, 23]. However, whether more advanced ideas in supervised AT can bring improvements in SAT is under-explored. In this section, we suggest that adversarial weight perturbation can be introduced into SSL pretext task for downstream robustness. Note that in the method proposed by Wu et al. [35], adversarial weight perturbation is applied to different supervised AT frameworks including vanilla PGD [25], TRADES [39], RST [3] and MART [34]. However, all those methods require full or partial labels and are based on classification loss.

The adversarial weight perturbation can be expressed as:

$$L_{\text{AWP}} = \max_{\hat{\theta} \in \mu(\theta)} L_{\text{CE}}(f_{\hat{\theta}}(x, y)) + \beta L_{\text{adv}}(f_{\hat{\theta}}(x^{\text{adv}}, y)), \tag{12}$$

where $\mu$ is the perturbation size of the model weight.

In contrast, in our work the weight perturbation is introduced into the SSL pretext task. Thus the perturbation doesn't regularize the weight classification-loss landscape, but works on the feature similarity loss in a label-free paradigm instead. This transition of learning paradigm makes it interesting to see whether such perturbations respect to the SSL objective can benefit downstream robust generalization. Specifically, the self-supervised weight adversarial perturbation perturbs the model weight in the direction of enlarging the self-supervised cosine similarity loss and increases the smoothness of this similarity loss landscape. The weight perturbation can be formulated as:

$$\hat{\theta}_2 = \arg \min_{\theta_2 \in \mu(\theta)} \text{Sim}(f_2(x_{\text{aug-weak}}), f_1(x_{\text{aug-weak}})) + \lambda \cdot \text{Sim}(f_2(x^{\text{adv}}_{\text{aug-weak}}), f_2(x_{\text{aug-weak}})). \tag{13}$$

The weight perturbation finds the "worst" adversarial scenario which is beneficial for model robustness. However, this extra adversarial component also further increases the difficulty of the task. In supervised AT, the existence of ground truth labels helps the model to converge despite the enlarged difficulty. However, both AT and SSL are difficult tasks. In the early stage of SAT, regulating the model to learn this rather difficult objective with respect to insufficiently learned adversarial features could impede the learning of natural features and we propose to apply this weight adversarial perturbation only in the later stage of learning when the pseudo-supervised learning of clean and adversarial features is stabilized. Without our weight self-perturbation scheme, there is a clean accuracy drop of 0.7% and PGD robust accuracy drop of 0.8% on CIFAR-10 with ResNet-34.

The adversarial weight perturbation calculated on weakly augmented data is combined with sample adversarial perturbations over the four sample distributions in the previous section. The overall learning objective is:

$$\ell_{swap-diverse-aug} = \sum_{aug_i} \ell_{clean} + \lambda \cdot \ell_{adv}, \tag{14}$$

where

$$\ell_{clean} = \text{Sim}(f_{\hat{\theta}_2-aug_i-clean}(x_{aug_i}), f_{\theta_1}(x_{aug_i})), \tag{15}$$

and

$$\ell_{adv} = \text{Sim}(f_{\hat{\theta}_2-aug_i-adv}(x^{\text{adv}}_{aug_i}), f_{\hat{\theta}_2-aug_i-clean}(x_{aug_i})). \tag{16}$$

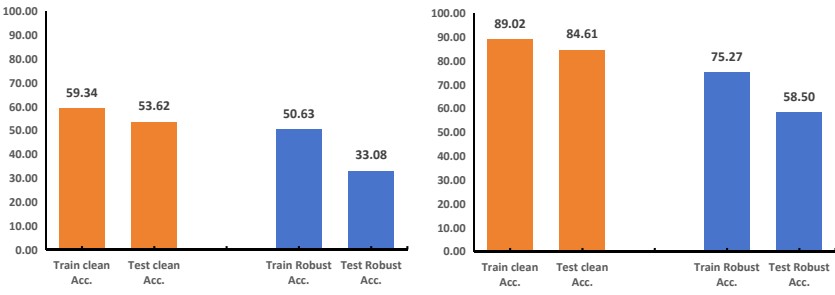

Figure 3: The generalization gap of SAT (SimCLR) with our proposed DAQ-SDP. The left part is the result on CIFAR-100. The right part is the result on CIFAR-10.

As shown in Figures 1 and 3, the robust generalization gap is reduced by around 3% and the test clean accuracy improves by more than 1.5% on average with our proposed method, which means properly regulating the smoothness of weight SSL-loss landscape in pre-training can improve the robust generalization of downstream classification despite the lack of labels.

### 4.3 Towards Unified Understanding of SAT and supervised AT

Despite the task difference between the learning paradigms, our method steps forward to an unified understanding of SAT and supervised AT by revealing their similar characteristics with respect to

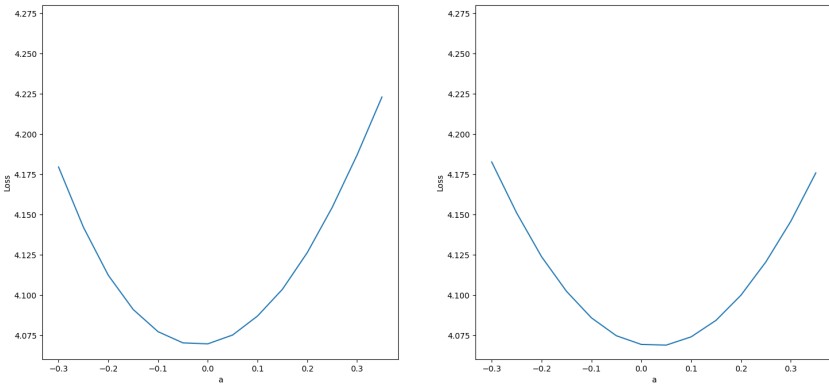

Figure 4: 1D visualization of the downstream weight loss landscape. The plot on the left is for the baseline method and the plot on the right shows that for our method with self-perturbed weight. The x-axis represents the magnitude to move the model weight.

Table 1: Results on ResNet-34 trained on CIFAR-10 with SimCLR framework.

| Evaluation | Method | Clean | PGD | AutoAttack |
|---|---|---|---|---|
| | DynACL[24]+AIR[37] | 79.79 | 51.07 | 47.61 |
| Simple Linear | TARO[23] | 84.23 | 53.36 | 45.68 |
| Finetuning | DecoupledACL[38] | 82.46 | 56.86 | 47.99 |
| | DAQ-SDP (ours) | **84.57** | **58.57** | **49.22** |

generalization. In this work, We also provide a "higher" level of perspective than previous narrower methods that focus on SAT with single SSL framework. We look forward to seeing future works with general approaches that improve adversarial training across different supervision settings.

## 5 Experiment

In this section, we demonstrate the effectiveness of our method. First, we evaluate the effectiveness when DAQ-SDP is plugged into contrastive and positive-pair only SSL frameworks. Then we conduct an ablation study for each part of our method. We also visualize the representation learned by our method through t-SNE [33] in Appendix.

Table 2: Results on ResNet-34 trained on CIFAR-100 with SimCLR framework.

| Evaluation | Method | Clean | PGD | AutoAttack |
|---|---|---|---|---|
| | DynACL[24]+AIR[37] | 47.02 | 23.91 | 20.66 |
| Simple Linear | TARO[23] | 51.28 | 29.46 | 21.14 |
| Finetuning | DecoupledACL[38] | 51.44 | 30.68 | 21.31 |
| | DAQ-SDP (ours) | **53.54** | **33.09** | **23.42** |

**Experimental Setup:** We apply our method on top of SimCLR [4], SimSiam [5] and BYOL [17] to evaluate the performance improvements across SSL frameworks. We also conduct extensive experiments on ResNet-18, ResNet-34 and ResNet-50. We find that the improvements are less significant on the smaller model ResNet-18. The rationale may be that simply increasing data complexity for ResNet-18 by using one extra strongly-augmented view could decrease the training set clean accuracy from 54.45% to 52.78% on CIFAR-100, suggesting insufficient model capacity to distill richer information while fitting well on clean data. This is understandable as model capacity takes a significant role in adversarial robustness and many techniques in supervised AT [30, 1, 35] require models with sufficient capacity to be effective. Given the complexity of AT and SSL, we expect a model larger than ResNet-18 is needed to learn rich information from the training data.

All SSL models in our method are first trained with clean data for 1000 epochs, then adversarially trained with 5-step PGD attack with the epsilon size of 8/255. Methods in previous works are adversarially trained for 1000 epoches as mentioned in their papers. The robustness is evaluated with

Table 3: Results on other SSL frameworks with ResNet-34 backbone. Note that most previous works are based on SimCLR and can not be used in Positive-Pair only SSL frameworks. The dataset is CIFAR-10.

| SSL Framework | Method | Clean | PGD | AutoAttack |
|---|---|---|---|---|
| SimSiam | TARO[23] | **81.71** | 52.61 | 44.46 |
| SimSiam | DecoupledACL[38] | 78.40 | 57.17 | 47.20 |
| SimSiam | DAQ-SDP (ours) | 80.42 | **58.53** | **47.69** |
| BYOL | TARO [23] | **86.84** | 52.01 | 44.76 |
| BYOL | DecoupledACL[38] | 83.15 | 55.22 | 47.67 |
| BYOL | DAQ-SDP (ours) | 85.91 | **56.53** | **49.07** |

Table 4: Results on ResNet-18 trained on CIFAR-10 with SimCLR framework.

| Method | Clean | PGD | AA |
|---|---|---|---|
| DynACL[24]+AIR[37] | 78.08 | 49.12 | 45.17 |
| TARO[23] | **82.86** | 52.44 | 43.99 |
| DecoupledACL[38] | 80.17 | 53.95 | **45.31** |
| DAQ-SDP(ours) | 81.76 | **55.15** | 45.12 |

AutoAttack [7] and PGD attack with 20 iterations and epsilon size of 8/255. $\lambda$ is set to 2. We use double adversarial perturbation after 60 epochs of training and weight perturbation size constraint of 0.002. The SLF and AFF finetuning details are the same as previous works [14, 38] with 25 steps of training and initial learning rate of 0.1. The experimental results in our method is the average of 5 runs, with a maximal variation range of $\pm 0.5$ for clean accuracy and $\pm 0.35$ for robust accuracy. All experiments are conducted on 2 RTX 3090 GPUs.

## 5.1 Effectiveness of DAQ-SDP across SSL Frameworks, Models and Datasets

a) We first conduct experiments with different SSL frameworks on ResNet-34. As shown in Table 1 and Table 2, our method contributes to significant improvements on both clean and robust accuracy on both CIFAR-100 and CIFAR-10 with SimCLR [4]. Note that TARO [23] is an adversarial sample generation method that needs to be combined with specific SAT baselines. In this work we combine it with the same baseline framework we use for better performance and fair comparison. Although TARO [23] gives slightly better clean accuracy on CIFAR-10, our method outperforms TARO [23] on robust accuracy by a large margin. On CIFAR-100, both our clean and robust accuracy outperforms TARO [23]. Also note that methods except TARO [23], DecoupledACL [38] and ours are contrastive based methods and don't generalize to positive-pair only SSL frameworks. In Table 3, we compare our method with previous works on SimSiam [5] and BYOL [17]. As shown in the results, our work provides a consistent improvements for different SSL frameworks. This is because we treat differently trained models as the teacher that provides supervision for the clean data feature space. Once we obtain the supervision, we take an SSL-framework agnostic process for robustness improvements.

b) We then conduct experiments on ResNet-18 and ResNet-50. From Table 4 and Table 5, our method shows improvement across model sizes.

c) We also provide cross-dataset transfer learning results. Table 6 shows transfer learning from CIFAR-100 to CIFAR-10, in which our work outperforms other SAT methods. In Table 7, we provide the transfer learning SLF results from CIFAR-10 to STL-10, which shows that our method can transfer well across datasets from more different domains.

Table 5: Results on ResNet-50 trained on CIFAR-10 with SimCLR framework.

| Method | Clean | PGD | AA |
|---|---|---|---|
| DynACL[24]+AIR[37] | 80.67 | / | 47.56 |
| TARO[23] | 84.57 | 53.60 | 46.86 |
| DecoupledACL[38] | 83.32 | 55.70 | 48.24 |
| DAQ-SDP(ours) | **85.22** | **58.05** | **49.49** |

Table 6: Cross-dataset transfer learning from CIFAR-100 to CIFAR-10. Note that the methods compared here are not restricted to specific SSL framework. We use both simple linear finetuning (SLF) and adversarial full finetuning (AFF) in this experiment. We use ResNet-34 as the backbone model.

| | SLF | | AFF | |
|---|---|---|---|---|
| Method | Clean | PGD | Clean | PGD |
| DecoupledACL[38] | 55.03 | 25.78 | 86.16 | 52.97 |
| TARO[23] | 57.13 | 23.99 | 86.00 | 52.71 |
| DAQ-SDP (ours) | **57.66** | **26.83** | **86.83** | **53.08** |

Table 7: Cross-dataset transfer learning from CIFAR-10 to STL-10. We use ResNet-34 as the backbone model.

| Method | Clean | PGD |
|---|---|---|
| Baseline | 63.84 | 40.66 |
| DAQ-SDP(ours) | **66.79** | **40.75** |

Table 8: Ablation study for our method with SimCLR on CIFAR-100. We use ResNet-34 as the backbone model.

| Method | Clean | PGD |
|---|---|---|
| Baseline | 51.44 | 30.68 |
| DAQ(single-BN) | 52.27 | 31.56 |
| Diverse Augmented Query | 52.67 | 32.11 |
| Weight Self-Perturbed Scheme | 51.56 | 32.37 |
| DAQ-SDP(ours) | **53.54** | **33.09** |

## 5.2 Ablation Study

In this section we evaluate each part of our method. Table 8 shows the results on CIFAR-100. With our Diverse Augmented Query, the clean accuracy increases by 1.23% and the PGD robust accuracy increases by 1.43%. With our Diverse Augmented Query and Weight Self-Perturbed Scheme, the clean accuracy increases by 2.10% and the robust accuracy increases by 2.41% . As shown in Figures 1 and 3, both robust generalization and clean accuracy are improved compared to the baseline. These improvements brought by our proposed method demonstrate the effectiveness of the enhanced sample complexity and model regularization with the SSL loss term.

The smoothness of the weight loss landscape has been shown to be important for robust generalization in supervised AT [35]. In Figure 4 we also analyze the effect of our method to downstream weight loss landscape through 1D visualization as previous work in supervised AT [35] did. From Figure 4 we can see that the regularizing effect transferred to downstream loss landscape is actually much attenuated compared with directly regularizing classification loss in supervised AT [35], showing difficulty of such a transferred improvement from SSL pretext tasks.

## 6 Conclusion

In this paper, we observe that SAT has the similar problem of large robust generalization gap and clean accuracy degradation as in supervised AT. We then propose a general method to solve this problem, which can be directly combined with different pretrained SSL models without further changing learning objectives. We first challenge the previous conclusion that diverse and strong augmentations harms SAT and propose a diversely augmented query based method with Aug-Adv Pairwise-BatchNorm to distill generalizable and diverse information from clean model. Second, different from previous works that focused on introducing sample perturbation to the SSL pretext task, we suggest that regulating the smoothness of the SSL loss landscape by adversarial weight self-perturbation boosts robust generalization transferable to downstream classification. Our method not only improves the performance across different SSL frameworks, but also provides insights for narrowing the gap between the study of these two adversarial learning paradigms.

## Acknowledgements

This research work is supported by the Project of Chinese Academy of Sciences (E141020).

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

# A    Appendix / supplemental material

## A.1    Feature Visualization

In this section, we visualize the the learned features of the test set of CIFAR-10 with t-SNE[33]. Each data point is colored with its label. As shown in 5, the features of different classes learned by our DAQ-SDP has a clearer boundary than the baseline method.

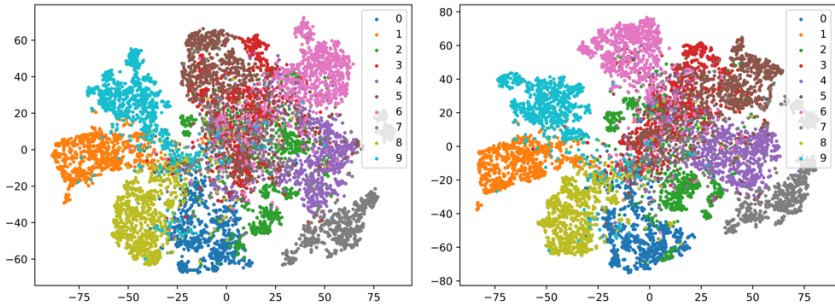

Figure 5: T-SNE results of the test set features. The left part of the figure is the features predicted by our adversarial training baseline and the right part of the figure is the features predicted by our DAQ-SDP.

## A.2    Societal Impacts

Our work is useful for pretraining robust models with no labels. However, it generated one more adversarial data and perturbed weights, which takes more computation. So it can cause more consumption of energy and pollution. Despite this limitation, we believe our method is still beneficial for the society and promotes model robustness in real life.

