# OpenReview forum: "Self-Supervised Adversarial Training via Diverse Augmented Queries and Self-Supervised Double Perturbation"
_NeurIPS.cc/2024/Conference — NeurIPS 2024 poster_

### Official Review · Reviewer_iqHN · 2024-07-01

**Soundness:** 3
**Presentation:** 3
**Contribution:** 3
**Rating:** 5
**Confidence:** 4

**Summary:**

The authors aim to further close the gap between the clean generalization gap and the robust generalization gap for SSL models. Several engineering improvements have been made to the SAT pipeline, such as including strong data augmentation, adversarially perturbing weight, and using separate BN. The resulting algorithm exhibits non-trivial improvements on benchmark datasets.

**Strengths:**

(1) The paper is overall well written. I find it easy to follow the motivations and implementations of the components.

(2) The argument about the usage of strong data augmentation is interesting and supported by experimental results.

(3) The performance gain seems non-trivial. An average improvement of 1% in robust accuracy can be expected.

**Weaknesses:**

Major:

(1) Baseline + Weight Self-Perturbed Scheme should be included in Table 4.

(2) The authors claim that " diversely augmented samples help SAT given that the model has sufficient capacity". I am wondering whether the difference between ResNet-18 (used in previous papers) and ResNet-34 (used in this paper) is large enough for the claim to hold. How about using even larger models, like ResNet-101 or Wide-ResNet-28-10?

(3) Also, I believe the ResNet-18 results should not be entirely excluded. Previous works only tuned their hyper-parameters on ResNet-18, while the hyper-parameters for the proposed algorithm are tuned on ResNet-34. The comparison may not be fair.

(4) The claims made in Section 4.3 are not well supported by the results, especially the "unified understanding of SAT and AT". I cannot see why applying AWP to SAT can lead to such "unified understanding".  Besides, "robustness and robust generalization can be transferable across SSL and classification" is also evidenced by previous works.


Minor:

(1) Figure 1 and Figure 3 seem blur. Are you using the JPEG images directly?

(2) Typo errors: the caption of Table 1. Equation 8.

**Questions:**

(1) How does the algorithm perform on ResNet-18?

**Limitations:**

See weakness.

---

> ### Author Rebuttal · Authors · 2024-08-07
>
> Dear reviewer iqHN,
>
> We appreciate your positive comments! Here is our response:
>
> 1. We added the results with ResNet18 and ResNet50 on CIFAR10 (larger model is hard for us to train in limited time). The results of ResNet18 are not included in our initial submission for it is not as obvious as in the larger model. The AIR[37] paper doesn’t include PGD accuracy on ResNet50 or checkpoint and it can take us a long time to train. So only clean and AA attack accuracy are provided.
>
> ResNet18:
> | Method | Clean | PGD | AA |
> |---|---|---|---|
> | DynACL+AIR| 78.08 | 49.12 | 45.17 |
> | TARO | 82.86 | 52.44 | 43.99 |
> |DecoupledACL| 80.17 | 53.95 | 45.31 |
> |DAQ-SDP(ours)| 81.76 | 55.15 | 45.12 |
>
> ResNet50:
> | Method | Clean | PGD | AA |
> |---|---|---|---|
> | DynACL+AIR| 80.67 | / | 47.56 |
> | TARO | 84.57 | 53.60 | 46.86 |
> |DecoupledACL| 83.32 | 55.70 | 48.24 |
> |DAQ-SDP(ours)| 85.22 | 58.05 | 49.49 |
>
> 2.	We added weight perturbation in the ablation study experiments:
> | Method | Clean | PGD |
> |---|---|---|
> |Baseline | 51.44 | 30.68 |
> |DAQ(single-BN) | 52.27 | 31.56 |
> |Diverse Augmented Query | 52.67 | 32.11 |
> |Weight Self-Perturbed Scheme | 51,56 | 32.37 |
> |DAQ-SDP(ours) | 53.54 | 33.09 |
>
> 3.	We added transfer learning results from CIFAR10 to STL 10:
> | Method | Clean | PGD |
> |---|---|---|
> |Baseline | 63.84 | 40.66 |
> |DAQ-SDP(ours) | 66.79 | 40.75 |
>
> 4.	We hope to argue that comparison between our method and previous ones on ResNet34 are fair. First, DynACL[24]+AIR[37] results are the same as reported in the paper. Second, DecoupledACL[38] hyperparameters on larger models are finetuned for clean-robust trade-off. The hyperparameter to tune is weight decay in adversarial training. On CIFAR100, weight decay of 2e-4 gives 51.44% (clean) and 30.68% (PGD) but a larger regularization with weight decay of 5e-4 gives 49.94% and 31.99%. On CIFAR10, weight decay of 5e-4 gives 82.46 (clean) and 56.86% (PGD) but a larger regularization of 1e-3 gives 80.57% and 56.48%. TARO[23] is an attack method that can be combined with different frameworks. The initial one-stage training results reported in the paper is much lower and we combined it with our model for improved results. The hyperparameter that matters is the batch size for choosing target to attack. We use the same batch size of 512. On ResNet50, the hyperparameter is finetuned in the same manner. An increase of weight decay from 5e-4 to 6e-4 on CIFAR10 typically causes a robustness reduction of around 1.00% in these methods.
>
> 5.	We are glad to discuss your questions about the claim of “unified perspective” and “robust generalization”. Indeed, the whole adversarial self-supervised learning literature is about robustness and generalization transferable to downstream tasks. And we hope to clarify the following points:
>
> a)	There are actually two types of generalization in the self-supervised AT context. The first is the generalization between tasks from the SSL perspective. For instance, SSL robustness transferring to other datasets and downstream classification tasks. At least to the best of our knowledge, all previous works explore the extent of this generalization. In this work we are actually curious about the robust generalization of SSL methods from the adversarial training (training-testing) perspective, which is related to the robust overfitting concept in supervised AT. Since SSL operates on features and needs an extra linear finetuning for classification loss, this generalization is often overlooked in previous works. Although we don’t observe decreased validation robustness during training (robust overfitting) as in supervised AT, we do observe a large robust generalization gap (in Figure 1 of the paper). Our method helps to solve this generalization problem (in Figure 3) and improves the results from this AT perspective for both contrastive and non-contrastive SSL (shown in the same dataset and cross-dataset downstream classification results). The related robust overfitting phenomenon is studied a lot in supervised AT but not studied in the SSL context. We believe the identification and analysis of such traits of self-supervised AT create a space for further improvements. That is what we mean by “providing a unified perspective across AT under different supervisions”. We look forward to your suggestions about making these phrases clearer.
>
> b)	In supervised AT, phenomenon of robust overfitting is prevalent and AWP helps to improve robustness. However, whether AWP works depends on the trade-off between reduced generalization gap and decreased training accuracy. Since there is a difference in the difficulty of supervised AT and self-supervised AT (which is “a challenging problem due to its two mixed challenging goals” and “clearly can be even more challenging than the semi-supervised AT setting” in [38]), it’s unclear whether the generalization gap for self-supervised AT is large enough to compensate for the trade-off for improvement. It’s also interesting that using the self-perturbed AWP scheme provides a 0.70% and 0.80% improvements over self-perturbed AWP for the whole training process on CIFAR10. Without the scheme there is a 0.88% improvement on PGD acc. but 0.81% reduction on clean acc. compared with the result without self-perturbed AWP. This can also be related to the traits of self-supervised AT.
>
> c)	Transferring preferrable AT properties except robustness itself, e.g. loss landscape smoothness, from SSL to classification is not considered in previous works and not as easy as one might think. In Figure 4 of the uploaded one-page pdf, we show the 1d visualization of resulting loss landscape of the baseline and our method. Although regularizing SSL objective smooths downstream loss landscape, the transferred regularization effect is much attenuated than directly regularizing classification loss as in [35], showing the difficulty of such an improvement.
>
> 6.	The figures are uploaded in the one-page pdf. Thanks again for your time and patience.

---

> > ### Comment · Reviewer_iqHN · 2024-08-08
> > **Thanks for the rebuttal**
> >
> > Dear authors,
> >
> > First I would like to express my gratitude for you to conduct the extra experiments, which I find to be helpful. However, there are two main concerns that I feel worrying after reading the rebuttal.
> >
> > ----
> >
> > **Academic Integrity**
> >
> > The authors state in the rebuttal that "The results of ResNet18 are not included in our initial submission for it is not as obvious as in the larger model.", which reveals that the authors have the intention to **exclude certain experiment results simply because they are not in favor of the authors' claim** rather than faithfully acknowledge the results as a limitation or give intuitive/rigorous explanation for the results in the initial submission.
> >
> > According to the NeurIPS Code of Conduct, "Scientific misconduct including fabrication, falsification, or plagiarism of paper submissions or research presentations, including demos, exhibits or posters.", I personally believe the authors' motivation of **hiding** the results on smaller models worrying.
> >
> > ----
> >
> > **Writing Clarity**
> >
> > The responses to Q4 and Q5 have no format, no highlight, and no summarization. Even after spending 15 minutes trying to understand the long paragraphs, I still cannot understand the core logic behind the authors' responses. Even though the original paper is written well, I still find the unclearness in the rebuttal unsatisfactory.
> >
> > ----
> >
> > For the above reasons, I have lowered my score to 3. I would like to reconsider my rating if the authors can provide more clarification.

---

> ### Author Response · Authors · 2024-08-08
> **response to question of reviewer iqHN**
>
> Dear reviewer iqHN,
>
> Thanks for your reply.
>
> First, we would like to argue that in the initial submission, we already pointed out that the performance of the method is more obvious on larger models than on ResNet18, that's because our method requires larger model capacity for better performance. **This claim is already included in the original submission.** You can find this from line 275 to line 284 in Experimental Setup section. **So we are not hiding results.** In the rebuttal, you require us to provide the results on other models, so we provided them.
>
> Second, the response to Q4 is trying to solve the worry that our experiments are unfair. We are arguing that **we finetuned the parameters on larger models and used appropriate batchsize.**
>
> Third, the response to Q5 is trying to explain our motivation. We first argued that **the generalization/overfitting problem from adversarial training perspective in self-supervised AT is overlooked** in previous works. But the identification of this problem creates a space for improvement. Then we argued that **the improvement brought by the self-perturbed adversarial weight technique is not that straightforward or easy.**
>
> Best Regards,
>
> Authors

---

> > ### Author Response · Authors · 2024-08-09
> > **further clarification for the rebuttal**
> >
> > Dear reviewer iqHN,
> >
> > We have updated the long paragraphs with new format, highlight and summary:
> >
> > In this paragraph, we hope to argue that **comparison between our method and previous ones on ResNet34 are fair** and **the hyperparameters are finetuned to our best effort or used appropriately**. **First**, DynACL[24]+AIR[37] results are the same as reported in their original paper. **Second**, DecoupledACL[38] hyperparameters on larger models are finetuned for clean-robust trade-off. The hyperparameter to tune is weight decay in adversarial training. On CIFAR100, weight decay of 2e-4 gives 51.44% (clean) and 30.68% (PGD) but a larger regularization with weight decay of 5e-4 gives 49.94% and 31.99%. On CIFAR10, weight decay of 5e-4 gives 82.46 (clean) and 56.86% (PGD) but a larger regularization of 1e-3 gives 80.57% and 56.48%. **Third**, TARO[23] is an attack method that can be combined with different frameworks. The initial one-stage training results reported in the paper is much lower and we combined it with our model for improved results. The hyperparameter that matters is the batch size for choosing target to attack. We use the same batch size of 512. On ResNet50, the hyperparameter is finetuned in the same manner. An increase of weight decay from 5e-4 to 6e-4 on CIFAR10 typically causes a robustness reduction of around 1.00% in these methods.

---

> > > ### Author Response · Authors · 2024-08-09
> > > **further clarification for rebuttal**
> > >
> > > We are glad to discuss **your questions about the claim of “unified perspective” and “robust generalization”** in this part. Indeed, the whole adversarial self-supervised learning literature is about robustness and generalization transferable to downstream tasks. But **there is a difference between previous works and ours**. And **in this part we hope to clarify the following points**:
> > >
> > > ---
> > >
> > > 1.	Since you have asked about the difference between our work and previous works that explored robustness and generalization in self-supervised AT. In this paragraph we hope to first explain  there are actually two types of generalization in the self-supervised AT context. **The first is the generalization between tasks from the SSL perspective**. For instance, SSL robustness generalizing to other datasets and downstream classification tasks. At least to the best of our knowledge, all previous works explore the extent of this generalization. In this work we are actually curious about **the robust generalization of SSL methods from the adversarial training (training-testing) perspective, which is related to the robust overfitting concept in supervised AT**. Since SSL operates on features and usually directly finetuned on downstream datasets, this generalization/overfitting during pretraining is often overlooked in previous works. This problem can be seen clearer if we conduct pretraining and finetuning on the train/test split of the same dataset. (This setting is called self-task transfer in previous works) Although we don’t observe decreased validation robustness during training (called **robust overfitting**) as in supervised AT, we do **observe a similar large robust generalization gap** (in Figure 1 of the paper). Our method **helps to solve this generalization problem** (in Figure 3) and **improves the results from this AT perspective for both contrastive and non-contrastive SSL** (shown in the same dataset and cross-dataset downstream classification results). The related robust overfitting phenomenon is studied a lot in supervised AT but not studied in the SSL context. **We believe the identification and analysis of such traits of self-supervised AT create a space for further improvements.** That is what we mean by “providing a unified perspective across AT under different supervisions”. We look forward to your suggestions about making these phrases clearer. $\newline$
> > >
> > > ---
> > >
> > > 2.	In this paragraph we would like to discuss **why our adversarial self-perturbed weight is not that trivial or straightforward and how is that related to our motivation**. In supervised AT, phenomenon of robust overfitting is prevalent and AWP helps to improve robustness because there is an obvious drop of validation robust accuracy in training. However, **whether AWP works depends on the trade-off between reduced generalization gap and decreased training accuracy.** Since there is a difference in the difficulty of supervised AT and self-supervised AT (which is “a challenging problem due to its two mixed challenging goals” and “clearly can be even more challenging than the semi-supervised AT setting” in [38]), **it’s not obvious whether the robustness generalization gap for self-supervised AT is large enough to compensate for the trade-off for improvement**. It’s also interesting that using the self-perturbed AWP scheme provides a 0.70% and 0.80% improvements over self-perturbed AWP for the whole training process on CIFAR10. Without the scheme there is a 0.88% improvement on PGD acc. but 0.81% reduction on clean acc. compared with the result without self-perturbed AWP. This can also be related to the traits of self-supervised AT.
> > >
> > > ---
> > >
> > > 3.	In this paragraph we further explain why is the improvement not that straightforward. **Transferring preferrable AT properties except robustness itself, e.g. loss landscape smoothness, from SSL to classification is not considered in previous works and not as easy as one might think**. In Figure 4 of the uploaded one-page pdf, we show the 1d visualization of resulting loss landscape of the baseline and our method. Although regularizing SSL objective smooths downstream loss landscape, the transferred regularization effect is much attenuated than directly regularizing classification loss as in [35], showing the difficulty of such an improvement.

---

> > > > ### Comment · Reviewer_iqHN · 2024-08-12
> > > > **Thanks for the rebuttal**
> > > >
> > > > Dear authors,
> > > >
> > > > First I would like to express my gratitude for you to add further clarification which I find to be helpful. However, there are three main concerns that I feel worrying after reading the continued rebuttal.
> > > >
> > > > ----
> > > >
> > > > **Academic Integrity**
> > > >
> > > > As a reviewer, I care more about your motivation behind leaving out the ResNet-18 result despite the model being adopted in most (if not all) previous works. How can the readers know the performance of your algorithm on smaller models? It is not ethical to exclude any important results, especially those related to the main claims, and write several unjustified claims to explain why you do so.
> > > >
> > > > ----
> > > >
> > > > **Writing Clarity**
> > > >
> > > > Reading sentences like these is pure pain:
> > > >
> > > > "the self-perturbed AWP scheme provides a 0.70% and 0.80% improvements over self-perturbed AWP for the whole training process
> > > >
> > > > - "a 0.70% and 0.80% improvements": This is not grammatically correct.
> > > >
> > > > - "self-perturbed AWP ... improve over ... self-perturbed AWP ...": Why are you comparing the same method?
> > > >
> > > > "The related robust overfitting phenomenon is studied a lot in supervised AT but not studied in the SSL context"
> > > >
> > > > - because SSL doesn't robustly overfit at all!
> > > >
> > > > ----
> > > >
> > > > **Writing correctness***
> > > >
> > > > The authors write that "non-contrastive methods BYOL and SimSiam".
> > > >
> > > > However, it appears that the primary reason that algorithms like BYOL and SimSiam work is that it is doing a form of contrastive learning—just via an indirect mechanism. I personally view them as contrastive learning with advanced tricks, as it is written by the SimSiam authors "so it (SimSiam) can also be thought of as 'SimCLR without negative pairs'".
> > > >
> > > > ----
> > > >
> > > > For the above reasons, I will keep my score at 3. I do not recommend acceptance.

---

> > > > > ### Author Response · Authors · 2024-08-13
> > > > >
> > > > > Dear reviewer iqHN,
> > > > >
> > > > > Thanks for your time and efforts in this discussion. We understand your worries about our motivation and writing clarity. We still really hope you can consider the following explanations:
> > > > >
> > > > > ---
> > > > >
> > > > > 1.	We humbly explain why **at least we didn’t deliberately hide the limitation of the method**. In the initial submission, we claimed that our method “helps SAT given that the model has sufficient capacity”, “ResNet18 … has insufficient model capacity to distill richer information” and “a model larger than ResNet18 is needed” for our method. By saying this, we tried to **make the main claim that the method has good performance on models larger than ResNet18**. All the experiments were used to support this point. **At least in our point of view**, **this description of our improvement contains no exaggeration**. Maybe this claim of limitation is still insufficient, but at least we didn’t intend to lead the reader to ignore this limitation. Otherwise, we would not repeat this claim several times in the paper. Actually, we mentioned this limitation during rebuttal again because we thought the limitation is already claimed and the claim in the paper is not exaggerated.
> > > > >
> > > > > ---
> > > > >
> > > > > 2.	We hope to argue that **BYOL and SimSiam were written as “non-contrastive”** in [41] (ICML 2021) and [23] (NIPS 2023). Both of these works analyze the difference between non-contrastive methods (specifically BYOL and SimSiam) and contrastive methods.
> > > > >
> > > > > ---
> > > > >
> > > > > 3.	We realize that we wrote long rebuttal paragraphs that contain too many unnecessary details and cause confusion. We feel very sorry for this. If possible, we hope to **give a very brief explanation for the clarity problem**:
> > > > > - In “self-perturbed AWP …”, we are comparing **clean and PGD accuracy** of our weight perturbation scheme (**applying AWP after 60 epochs**) and applying weight perturbation for **the whole training process**.
> > > > > - As you point out, the “robust overfitting” defined as “reduction of test accuracy during adversarial training” in supervised AT is not observed in SAT. However, at least based on our experimental result, there still exists a related **training-testing robust generalization gap problem**. This paper is trying to solve this problem.
> > > > >
> > > > > ---
> > > > >
> > > > > We thank again for your efforts in reviewing our paper. Your suggestions about writing clarity and formatting paragraphs are very helpful. We really appreciate it.
> > > > >
> > > > > Best Regards,
> > > > >
> > > > > Authors
> > > > >
> > > > > [23] Minseon Kim, Hyeonjeong Ha, Sooel Son, and Sung Ju Hwang. Effective Targeted Attacks for Adversarial Self-Supervised Learning. In NeurIPS, pages 56885–56902, 2023.
> > > > >
> > > > > [41] Yuandong Tian, Xinlei Chen and Surya Ganguli. Understanding Self-Supervised Learning Dynamics without Contrastive Pairs. In ICML 2021.

---

> > > > > > ### Comment · Reviewer_iqHN · 2024-08-13
> > > > > > **Thanks for the rebuttal**
> > > > > >
> > > > > > Dear authors:
> > > > > >
> > > > > > I acknowledge your diligent effort during the rebuttal period. After reading the latest rebuttal as well as comments from other reviewers, I believe my main concerns are mostly resolved.
> > > > > >
> > > > > > I have adjusted my rating accordingly and I hope that the extra experiments, discussion, and clarification will be incorporated into the revised version.
> > > > > >
> > > > > > All the best

---

> > > > > > > ### Author Response · Authors · 2024-08-14
> > > > > > >
> > > > > > > Dear reviewer iqHN,
> > > > > > >
> > > > > > > Thanks for your reply. We will incorporate all the suggestions from reviewers into the revised version.
> > > > > > >
> > > > > > > Best Regards.

---

### Official Review · Reviewer_6QGn · 2024-07-10

**Soundness:** 2
**Presentation:** 1
**Contribution:** 2
**Rating:** 5
**Confidence:** 3

**Summary:**

The paper introduces the DAQ-SDP (Diverse Augmented Queries Self-supervised Double Perturbation) method to solve the problem of large robust generalization gap and clean accuracy degradation in self-supervised adversarial training. The experimental results demonstrate the effectiveness of the DAQ-SDP method.

**Strengths:**

1. The experimental results show that DAQ-SDP is better than other baselines such as DecoupledACL and TARO.
2. Unlike most works that only perturb samples, this paper introduces the weight perturbation into the SSL pretext task.

**Weaknesses:**

1. Assumptions lack reasonable basis. (In line 56)
2. There are some typos:
	- Incorrect citation and reference:
		- Using \citet is better. (In lines 39, 122, 132, etc.）
		- In lines 314 and 315, "Figure 1" is linked to Table 1.
	- Inconsistent names:
		- "Pairwise-BatchNorm" (in line 63) and "pairwise BatchNorm" (in line 209)
		- "resNet34" (in caption of Table 1) and "ResNet34" (in caption of Table 2)
		- "Taro" (in lines 297, 300, 301, 302) and "TARO" (in Tables 1, 2, 5)
	- Some typos:
		- "clean" instead of "lean" in Eq. 8.
		- "note" instead of "Note" in Line 302.
		- "where" instead of "Where" in Line 234.
		- "a result" ? in Line 308.
3. Figures seem to be a bit fuzzy.

**Questions:**

1. Could you share the reason why you chose these hyperparameters (In lines 284-288) and whether you have done experiments with other hyperparameters?
2. How does DAQ-SDP compare to DecoupledACL and TARO in terms of runtime?
3. Can other models be used to prove the effectiveness of the approach?

**Limitations:**

The author mentioned that this approach is only applicable to a sufficiently large model (ResNet34).

---

> ### Author Rebuttal · Authors · 2024-08-07
>
> Dear reviewer 6QGn,
>
> We appreciate your efforts and time in reviewing our paper. The following is our response.
> First, the assumption in line 56 is also supported by [38]. This work includes the following description for self-supervised adversarial learning:
>
> 1.	“A challenging problem due to its two mixed challenging goals”.
> 2.	“Clearly … can be even more challenging than the semi-supervised AT setting.”
>
> SSL models typically require 1000 epochs to train. In the adversarial training of SSL models, no information about decision boundary is available and the features are required to be robust against attacks towards any directions away from its original place. That explains why  we describe them as hard tasks in our paper.
>
> Second, we choose the hyperparameters in line 284 to 288 for the following reasons:
>
> 1.	The training epochs, attack steps, attack range in training and attack range in testing are the standards reported in a series of previous works including [22][20][14][38][37].
>
> 2.	We conducted experiments and finds that using the 60-epoch self-perturbed AWP scheme provides a 0.70% and 0.80% improvements over self-perturbed AWP for the whole training process on CIFAR10. Without the scheme there is a 0.88% improvement on PGD accuracy but 0.81% reduction on natural accuracy compared with the result without self-perturbed AWP.
>
> 3.	We choose this weight self-perturbation range because it’s in the middle of the perturbation range used in supervised adversarial learning work [35] to give a good trade-off on robust generalization gap and training accuracy. We also show the effectiveness of our method with this hyperparameter with the experimental improvements. Please check the following ablation study results:
>
>
> Third, we have added the experiments on ResNet18 and ResNet50. The results of ResNet18 are not included in our initial submission for the result is not as obvious as in the larger model case. Please note that the DYNACL[24]+AIR[37] paper doesn’t include the PGD accuracy on ResNet50 or checkpoint and it take us too long to train for this result. So only clean and AA attack accuracy are provided. The experiments are conducted on CIFAR10.
>
> ResNet18:
> | Method | Clean | PGD | AA |
> |---|---|---|---|
> | DynACL+AIR| 78.08 | 49.12 | 45.17 |
> | TARO | 82.86 | 52.44 | 43.99 |
> |DecoupledACL| 80.17 | 53.95 | 45.31 |
> |DAQ-SDP(ours)| 81.76 | 55.15 | 45.12 |
>
> ResNet50:
> | Method | Clean | PGD | AA |
> |---|---|---|---|
> | DynACL+AIR| 80.67 | / | 47.56 |
> | TARO | 84.57 | 53.60 | 46.86 |
> |DecoupledACL| 83.32 | 55.70 | 48.24 |
> |DAQ-SDP(ours)| 85.22 | 58.05 | 49.49 |
>
> The epoch time for the baseline, diverse augmented query and weight self-perturbation in the later epochs is 2min52s, 5min24s and 6min29s respectively. We have included a new Social Impact section for this limitation.
>
> We are sorry that the figures are fuzzy and reuploaded the Figure 1 and Figure 3 in the one-page pdf. Also we have corrected the typos including reference, consistent method name and “the result” in line 308. We will change \citep to \citet in our new version.
>
> Thanks for your helpful advice. You can also check our explanation about the motivation in other responses.
> We appreciate your efforts and time. Thank you!

---

> ### Author Response · Authors · 2024-08-12
>
> Sorry for missing the ablation study results in the previous reply. The ablation study result on CIFAR100 with SimCLR is here:
> | Method | Clean | PGD |
> |---|---|---|
> |Baseline | 51.44 | 30.68 |
> |DAQ(single-BN) | 52.27 | 31.56 |
> |Diverse Augmented Query | 52.67 | 32.11 |
> |Weight Self-Perturbed Scheme | 51.56 | 32.37 |
> |DAQ-SDP(ours) | 53.54 | 33.09 |

---

> > ### Author Response · Authors · 2024-08-12
> >
> > Please also note that although the results with ResNet18 and ResNet50 are not included the original submission, we have clearly stated the limitation of the method with respect to model size from line 275 to line 284 in the Experimental Setup section.

---

> ### Author Response · Authors · 2024-08-12
>
> We further clarify the runtime of DecoupledACL, TARO and our method DAQ-SDP:
> DecoupledACL takes 2min52s per epoch. TARO takes 2min54s per epoch. Our method DAQ-SDP takes 5min24s in the first 60 epochs and 6min29s in the last 40 epochs.

---

> > ### Comment · Reviewer_6QGn · 2024-08-13
> >
> > Thanks for your response. My concerns have been solved. I raised my score.

---

### Official Review · Reviewer_dErS · 2024-07-12

**Soundness:** 2
**Presentation:** 1
**Contribution:** 2
**Rating:** 5
**Confidence:** 5

**Summary:**

The paper proposes a method to improve self-supervised adversarial training. This method consists of two stages. First, a standard self-supervised model is trained on clean images to learn a feature extractor network F-1. In the second stage, a robust feature extractor F_2 is trained based on the features generated by F_1. The total loss is a linear combination of standard self-supervised loss, where F-2 learns to mimic the features generated by F_1, and an adversarial loss. The adversarial loss follows adversarial training framework where the targets are provided by F_1. The method applies weak and strong augmentations in both stages. However, in order to process different streams of data, the method uses four sets of batch-norm layers for weak and strong augmentations and adversarial and clean images. Moreover, the method applies adversarial weight perturbation for downstream robustness.

**Strengths:**

The paper contextualizes well within the literature. The method brings a combination of current ideas from supervised adversarial training to self-supervised adversarial training.

**Weaknesses:**

The proposed approach does not include substantial novelty, where the main idea is following the two stage adversarial training in [38]. The addition of weight perturbation from [35] does not prompt a creative combination of ideas.

The presentation clarity can be substantially improved. There are several terms that are never defined in the equations such as aug_i and  f2_augi_clean in eqs 9,10,11, and g and h in lines 144, 150. In eqs 3 and 4 l_CL is a three-variable function, whereas it is defined as a two-variable function in eq 1. There are several typos such as line 171 "we aims", 185 "In the filed of", 242 "loss landscape respect".

The claim of unifying self-supervised and supervised adversarial training in section 4.3 is not supported empirically or theoretically.

**Questions:**

There are no experiments to show the outcome of weight perturbation on the loss landscape of the proposed method. Overall the paper is missing qualitative experiments.

**Limitations:**

The authors discuss aspects of limitations of their method in introduction, however potential negative societal impact is not included.

---

> ### Author Rebuttal · Authors · 2024-08-07
>
> Dear reviewer dErS,
>
> We appreciate your time and efforts in reviewing our paper. We are sorry the phrases of “unified perspective” and “unified understanding” cause confusion. First please let us explain the following points:
>
> a)	There are actually two types of generalization in the self-supervised adversarial training context. The first is the generalization between tasks from the SSL perspective. For instance, SSL robustness transferring to other datasets and downstream classification tasks. At least to the best of our knowledge, all previous works explore the extent of this generalization. In this work we are curious about the robust generalization of SSL methods from the adversarial training (training-testing) perspective, which is related to the robust overfitting concept in supervised AT. Since SSL operates on features and needs an extra linear finetuning for classification loss, this overfitting is often overlooked in previous works. Although we don’t observe decreasing validation robustness during training (robust overfitting) as in supervised adversarial training, we do observe a large robust generalization gap (shown in Figure 1). Our method helps to solve this generalization problem (in Figure 3) and improves the results from this AT perspective for both contrastive and non-contrastive SSL methods (shown in the same-dataset or cross-dataset finetuned results). The robust overfitting phenomenon is studied a lot in supervised AT but not studied in the SSL context. We believe the identification and analysis of such traits of self-supervised AT create a space for further improvements. That is what we mean by “bridging the gap and providing a unified perspective for adversarial training under different supervisions”. We look forward to your suggestions about making this phrase clearer.
>
> b)	In supervised AT, the phenomenon of robust overfitting is prevalent and AWP helps to improve robustness. However, whether AWP works depends on the trade-off between generalization gap and decreased training accuracy. Since there is a difference in the difficulty of supervised AT and self-supervised AT (which is “a challenging problem due to its two mixed challenging goals” and “clearly even more challenging than the semi-supervised AT setting” in [38]), it is unclear whether the generalization gap for self-supervised AT is large enough to compensate for the trade-off to achieve robustness improvement and transferred to classification.
>
> Second, we add some additional experiments.
> a)	Tables below show results on ResNet18 and ResNet50 with SimCLR on CIFAR10：
>
> ResNet18:
> | Method | Clean | PGD | AA |
> |---|---|---|---|
> | DynACL+AIR| 78.08 | 49.12 | 45.17 |
> | TARO | 82.86 | 52.44 | 43.99 |
> |DecoupledACL| 80.17 | 53.95 | 45.31 |
> |DAQ-SDP(ours)| 81.76 | 55.15 | 45.12 |
>
> ResNet50:
> | Method | Clean | PGD | AA |
> |---|---|---|---|
> | DynACL+AIR| 80.67 | / | 47.56 |
> | TARO | 84.57 | 53.60 | 46.86 |
> |DecoupledACL| 83.32 | 55.70 | 48.24 |
> |DAQ-SDP(ours)| 85.22 | 58.05 | 49.49 |
>
> b)	This table shows ablation study with self-perturbed weight：
>
> | Method | Clean | PGD |
> |---|---|---|
> |Baseline | 51.44 | 30.68 |
> |DAQ(single-BN) | 52.27 | 31.56 |
> |Diverse Augmented Query | 52.67 | 32.11 |
> |Weight Self-Perturbed Scheme | 51,56 | 32.37 |
> |DAQ-SDP(ours) | 53.54 | 33.09 |
>
> c)	This table shows transfer learning from CIFAR10 to STL10：
> | Method | Clean | PGD |
> |---|---|---|
> |Baseline | 63.84 | 40.66 |
> |DAQ-SDP(ours) | 66.79 | 40.75 |
>
> d)	Figure 4 in the newly uploaded pdf shows the effects of baseline and our method with self-perturbed weight on downstream weight loss landscape with 1D visualization. The regularizing effect transferred to downstream loss landscape is actually much attenuated compared with regularizing classification loss as in [35], showing difficulty of such a transferred improvement.
>
> Third, thanks for pointing out the clarity problem. $CL$ loss in eq1 is defined as a two-variable function and $ACL$ loss in eqs 3 and 4 is defined as a three-variable because of the added adversarial view of data. Different works have slightly different way to deal with this extra variable. For instance, [14] averages the two-variable version. Projector $g$ is a component that preserves instance discriminative feature. Predictor $h$ is a component to prevent model callapse. $aug_i$ represents differently augmented data. $f_{2-aug_{i}-clean}$ means the parameters of the student model with the pairwise-BN decided by augmentation strength and adversarial type.
>
> We respectfully argue that lots of previous works have a simple but effective idea. For instance, [14] combines pseudo-supervised classification, an idea used in semi-supervised AT [26], and high frequency component, a technique shown to be effective for robustness in [41], with ACL. [18] has a simple idea that similar features should be removed from negative pairs. [24] and [38] has the simple idea that AT and SSL should have separate hyperparameters. Even the first ACL works [20][22] are based on an idea that sounds straightforward: if the feature cannot be perturbed to be far from its original place, then the feature is robust. This is actually a simple combination of the notion of adversarial attack and instance discrimination. Identifying generalization problem in Self-Supervised AT and adjusting traits of downstream loss landscape via regularizing pretext tasks is actually not straightforward in the way it sounds.
>
> We have included the Societal Impact section:
> Our work is useful for pretraining robust models with no label. However, it generates one more adversarial data and perturbed weight, which take more computation. So it can cause more consumption of energy and pollution. Despite this limitation, we believe our method is still beneficial for the society and promotes model robustness in real life.
>
> [41] Haohan Wang, Xindi Wu, Zeyi Huang and Eric P. Xing. High-frequency Component Helps Explain the Generalization of Convolutional Neural Networks. In CVPR, 2020.

---

> > ### Comment · Reviewer_dErS · 2024-08-12
> >
> > I would like to acknowledge that I have read the authors' rebuttal. The new experiments and visualizations address some of my concerns. Thus, I increased my rating.

---

### Official Review · Reviewer_mcfF · 2024-07-13

**Soundness:** 3
**Presentation:** 3
**Contribution:** 3
**Rating:** 7
**Confidence:** 3

**Summary:**

This paper proposes a method to solve the robust generalization problem for self-supervised adversarial training in general. Starting by showing the generalization gap in existing self-supervised adversarial training framework, it proposes to solve the problem from the aspects of data complexity and model regularization and provides self-supervised remedies. This paper reduces the gap between the understanding for traditional adversarial training and self-supervised adversarial training by demonstrating the connection of their characteristics and helps to build a “unified perspective for adversarial training” under different scenarios. Robust accuracy and clean accuracy are improved across multiple self-supervised learning frameworks.

**Strengths:**

This is a paper with novelty and good quality overall:
1.It provides novel explorations and insights for the self-supervised adversarial learning paradigm, which has a shift of learning objective and different “task complexity” from the traditional supervised adversarial training. This paper can be regarded as the first work that analyzes the training problem of self-supervised adversarial learning across self-supervised frameworks in general as in the supervised scenario, while such a study is absent in previous researches. In the end, the connection with the study of traditional supervised adversarial training is built. I agree that this paper provides insight to help understand adversarial training under different supervision conditions in a broad view and has the potential to inspire future works in this field.
2.In the methodology level, the method proposed is adaptable and general. It’s mentioned in this paper that the method is easily plugged into the self-supervised learning frameworks or pretrained models for improvements. As the authors suggest, one advantage of the method is that it doesn’t require reformulating the learning objectives, finetuning complex hyperparameters or adversarially retraining the model from scratch.
The experiment result on multiple self-supervised frameworks shows consistent improvement.

**Weaknesses:**

1.To illustrate the usage of the “pairwise-BatchNorm” technique, the authors may need to provide ablation study that uses normal BatchNorm instead for comparison.
2.Some notations lack explanations in the paper. As an example, the notation related to theta are inconsistent in equation (10) and (11). Which theta in equation (10) is used in equation (11)?
3.There are typos or missing signs in the formulas, e.g. the adversarial data generation in equation (11).

**Questions:**

1.Correcting the typos in the formulas and explain the important notations are needed.
2.Additional ablation results for the “pairwise-BatchNorm” are needed.

**Limitations:**

The authors addressed the requirement for sufficient model capacity based on the complexity of this adversarial training task. This is acceptable. There is no negative societal impact.

---

> ### Author Rebuttal · Authors · 2024-08-07
>
> Dear reviewer mcfF,
>
> We appreciate your positive comments. Here is our response for your questions:
>
> 1. We have added new experiments on ResNet18 and ResNet50 to prove the effectiveness of our method.
>
> ResNet18:
> | Method | Clean | PGD | AA |
> |---|---|---|---|
> | DynACL+AIR| 78.08 | 49.12 | 45.17 |
> | TARO | 82.86 | 52.44 | 43.99 |
> |DecoupledACL| 80.17 | 53.95 | 45.31 |
> |DAQ-SDP(ours)| 81.76 | 55.15 | 45.12 |
>
> ResNet50:
> | Method | Clean | PGD | AA |
> |---|---|---|---|
> | DynACL+AIR| 80.67 | / | 47.56 |
> | TARO | 84.57 | 53.60 | 46.86 |
> |DecoupledACL| 83.32 | 55.70 | 48.24 |
> |DAQ-SDP(ours)| 85.22 | 58.05 | 49.49 |
>
> 2. We have added more ablation study including the BN technique.
> | Method | Clean | PGD |
> |---|---|---|
> |Baseline | 51.44 | 30.68 |
> |DAQ(single-BN) | 52.27 | 31.56 |
> |Diverse Augmented Query | 52.67 | 32.11 |
> |Weight Self-Perturbed Scheme | 51,56 | 32.37 |
> |DAQ-SDP(ours) | 53.54 | 33.09 |
>
> 3. We added some explanations for the notations in our paper:
> The $aug_i$ in the equations represents either strongly or weakly augmented data. $f_{2-aug_{i}-clean}$ and $f_{2-aug_{i}-adv}$ in equation 10 represent the parameters of the student model with the pairwise-BN decided by augmentation strength and adversarial category. Since the adversarial data generation in equation 11 works on the adversarial branch, the parameter used in equation 11 is $f_{2-aug_{i}-adv}$. $f_{2-aug_{i}-adv}$ is the same as $f_{\theta_2-aug_{i}-adv}$ in equation 15.
> 4. We have added additional Societal Impact section:
>
> Our work is useful for pretraining robust models with no label. However, it generates one more adversarial data and self-perturbed weight, which take more time and computation. So it can cause more consumption of energy and pollution. Despite this limitation, we believe our method is still beneficial for the society and promotes model robustness in real life.
>
> Thanks again for your positive comments!

---

### Author Rebuttal · Authors · 2024-08-07

We have uploaded a one-page pdf file containing unclear figures in the original submission and additional visualizations.

---

### Decision · Program_Chairs · 2024-09-25

**Decision:**

Accept (poster)

**Comment:**

This work analyzes and improves self-supervised adversarial training (SAT) as it relates to adversarial training (AT). The results improve across a variety of self-supervised schemes and the reviewer consensus is positive. Importantly this is an informative study at a higher "level" than prior narrower studies of SAT, in its proposal of a general approach across specific self-supervision methods, which can promote more research on adversarial training without supervision. The AC sides with acceptance.

Four reviewers with expertise in domain adaptation, self-supervision, distribution alignment, and adversarial vote for acceptance (mcfF: 7) and borderline acceptnace (dErS: 5, 6QGn: 5, iqHN: 5). The authors provide a rebuttal, all reviewers engage with the rebuttal, and the authors and reviewers discuss across multiple threads and posts.

To summarize, the main argument for acceptance is the improved understanding and improved empirical results for SAT relative to AT which are delivered by a general approach consisting of the novel coordination of data and model perturbations alongside the factorization of model statistics. While elements exist in prior work, they are appropriately credited, and the proposed method is sufficiently novel. Lastly the empirical findings counter past evaluations and therefore provide valuable signal. On the other hand, the weaknesses raised during review are (1) issues in notation and lacking ablation (mcfF), (2) questions of clarity (mcfF, dErS, 6QGn), (3) concern for novelty vs. two-stage AT (dErS), and (4) lacking coverage across model sizes like a smaller ResNet-18 in particular and more generally a question of limitations (iqHN). Each of the weaknesses was satisfactorily resolved by the rebuttal and discussion, and during the AC-reviewer discussion phase no additional issues were raised.

The productive discussion and clarifications led to higher scores for mcfF and 6QGn and iqHN. The thread with iqHN is significant, because it was the most negative, but it ultimately resulted in support of borderline acceptance. The area chair monitored this thread closely, and commends the authors for preserving a factual and respectful tone, and acknowledges the reviewer for the respectful and thoughtful conclusion of the thread.

Miscellaneous feedback:

- There is a typo in the abstract for "generation gap" vs. "generalization gap".
- This is a matter of style, but the number of significant digits in plots such as Figures 1 & 3 are perhaps distractingly high for the range of results, so consider lesser precision for presentation.